# Trends in Participation Rates of the National Cancer Screening Program among Cancer Survivors in Korea

**DOI:** 10.3390/cancers13010081

**Published:** 2020-12-30

**Authors:** E Hwa Yun, Seri Hong, Eun Young Her, Bomi Park, Mina Suh, Kui Son Choi, Jae Kwan Jun

**Affiliations:** 1National Cancer Control Institute, National Cancer Center, 323 Ilsan-ro, Goyang 10408, Korea; ehwayun@ncc.re.kr (E.H.Y.); srhong@ncc.re.kr (S.H.); dmsdudhey@ncc.re.kr (E.Y.H.); bmp0228@ncc.re.kr (B.P.); omnibus@ncc.re.kr (M.S.); kschoi@ncc.re.kr (K.S.C.); 2Graduate School of Cancer Science and Policy, National Cancer Center, 323 Ilsan-ro, Goyang 10408, Korea

**Keywords:** cancer, cancer screening, cancer survivor

## Abstract

**Simple Summary:**

Cancer survivors with a fear of cancer recurrence and a second, primary cancer after treatment take part in cancer screening as a self-care strategy. In other countries, the need for cancer screening guidelines for cancer survivors was identified by the participation rates of cancer screening among cancer survivors. However, there is no report of cancer screening practices among cancer survivors in Korea. Therefore, this study explored the participation rates of the cancer screening program among cancer survivors. The results of this study provide information on how many cancer survivors were screened for cancer and indicate the need for appropriate cancer screening guidelines for cancer survivors.

**Abstract:**

The study aimed to describe the participation rates of the National Cancer Screening Program (NCSP) among cancer survivors in Korea. The NCSP protocol recommends that all Korean men and women should be screened for cancer. Cancer survivors were defined as those registered for any cancer in the Korea Central Cancer Registry by December 31 of the year prior to being included in the target population of the NCSP. In this study, the participation rates for the NCSP were calculated as the percentage of people who participated in four kinds of cancer screening programs, independently. The average annual percentage change was assessed. The participation rates of the general population and cancer survivors were higher than 40% in stomach, breast, and cervical cancer screening. These rates were higher than that of colorectal cancer screening in 2014. In addition, the participation rates in the NCSP in 2002–2014 increased for all cancer types. The NCSP participation rates of the cancer survivors indicate the high demand for cancer screening. Further research may investigate the effect of the NCSP on second cancer occurrence or mortality in cancer survivors and the significance of cancer screening guidelines for cancer survivors.

## 1. Introduction

The growing population of cancer survivors has prompted an increase in attention to their unmet needs [1,2]. These include fear of cancer recurrence and progression, uncertainty about the future, and reduction of stress and pain [2,3,4,5]. Fear of cancer recurrence is one of the most frequently reported unmet needs [5]. Cancer incidence or the associated risk was high among cancer survivors compared with the general population. The second cancer incidence was reported to be approximately 8% among cancer survivors in the United States of America (USA) [6]. The 10-year risk of a second primary cancer among cancer survivors ranged from 6.2% to 44% [7]. A high incidence of a second cancer indicates the need for cancer screening among survivors to relieve their fear of recurrence. This also prolongs their survival through early detection and treatment of cancer recurrence or a new primary cancer occurrence.

In Korea, the five-year relative survival rate has improved from 42.9% in 1993–1995 to 70.4% in 2013–2017 [8]. This increasing trend has resulted in an increase in cancer prevalence, with 1,867,405 patients as of 1 January 2018 [8]. In many countries, cancer screening guidelines for the general population exist [9,10]; however, many other countries, including Korea, have no cancer screening guidelines for cancer survivors. In particular, the difference between the cancer screening guideline in Korea versus other countries is that stomach cancer screening is recommended because of the high incidence of stomach cancer in Korea. In 2017, stomach cancer was ranked the second most common cancer in Korea, following thyroid cancer [8].

In this context, cancer screening practices among cancer survivors have been reported in other countries [11,12,13,14]. However, there are no reports on cancer screening practices among cancer survivors in Korea. Therefore, this study described cancer screening practices among cancer survivors and compared cancer screening practices between cancer survivors and the general population in terms of income level and time since diagnosis.

## 2. Results

### 2.1. The Target Population of the National Cancer Screening Program

The target population for the last 13 years is shown in Table 1. The target population of the National Cancer Screening Program (NCSP), which varies by cancer screening type, has increased. The proportion of cancer survivors among the NCSP target population has also increased. The proportion of stomach cancer screening among the NCSP target population increased from 2.1% in 2002 to 5.9% in 2014. From 2004 to 2011, the proportion of cancer survivors among target populations of colorectal cancer screening increased from 3.4% to 6.1%. The recommended frequency for colorectal cancer screening was changed from every two years to one year in 2012. At that time, the total target population of the colorectal cancer screening program increased by approximately 1.5 times. In 2014, the proportion of cancer survivors among the NCSP target population of colorectal cancer screening was 7.5%. In addition, the proportion of cancer survivors has increased by about three times in the last 13 years for breast and cervical cancer screening.

### 2.2. Trends in the Participation Rates of NCSP

The trends in the participation rates in the NCSP among the general population and cancer survivors are presented in Table 2. Among the general population and cancer survivors, the participation rate in the NCSP from 2002 to 2014 increased for all cancer screening types. However, the magnitudes of this increase varied between cancer screening types. In 2014, the participation rates of the general population and cancer survivors were higher than 40% and were reported to be higher than that for colorectal cancer screening.

When examining the cancer screening participation rates by cancer types, the participation rates for cancer survivors and the general population were similar in 2014, except for those observed in breast cancer screening. In breast cancer, the participation rate for cancer survivors was 49.7%, while that of the general population was 54.3%. In 2002–2014, the participation rates of cancer survivors increased, and a similar trend was noted in the general population. However, the magnitude of the increasing trend in participation rates among cancer survivors was higher than that of the general population in cervical cancer screening. For cervical cancer, the annual percent changes (APCs) were 7.2% (95% confidence interval (CI), 4.3–10.1) and 4.0% (95% CI, 0.9–7.2) in the general population.

The NCSP participation rate for each cancer varied with economic status. However, it has increased in the last 13 years in both the general population and cancer survivors (Table 3). In 2014, the lowest NCSP participation rate was observed among the general population in the lower economic status group. This observation was consistent in cancer survivors as well. In addition, the cancer screening rates in both populations were similar. However, in terms of the participation rates among cancer survivors, in 2002–2014 the highest rate was observed in the lower economic status group for all cancer screening types.

### 2.3. Trends in the Participation Rates of NCSP by Group Based on Time Since Diagnosis

Based on the time since diagnosis, there was a consistently increasing trend in the NCSP participation rates among cancer survivors for over 10 years (Figure 1). The participation rates of cancer survivors were lowest at less than one year after diagnosis. Meanwhile, cancer survivors in all cancer screening types, except for cervical cancer, had the highest participation rate at more than five years after diagnosis. Until 2009, the participation rate of cervical cancer screening in the NCSP was similar to that of other cancer screening. After 2009, the participation rate of cervical cancer screening was highest at three to five years after diagnosis and lowest at less than one year after diagnosis.

## 3. Discussion

In Korea, the target population size of the NCSP has increased from 2002 to 2014. The proportion of cancer survivors in the NCSP target population has also increased over the last 13 years. One possible reason for the increased NCSP target population is its criteria, which sets a minimum age for screening, but lacks an upper age limit [15]. Another reason is the aging Korean population [16]. In Korea, the participation rate in the NCSP has increased over the last 13 years in both cancer survivors and the general population. Even without a cancer guideline for cancer survivors, the magnitude of the increased NCSP participation rate was high in the low economic status group among cancer survivors. Overall, based on the time since diagnosis, the NCSP participation rates among cancer survivors were high at more than five years after diagnosis. This study was based on insurance data. It presented trends in the participation rate of stomach cancer screening among cancer survivors for the first time in the world.

Most of the cancer screening practices for cancer survivors in the USA, United Kingdom, and Canada were reported in cross-sectional surveys with self-reported or insurance data. The reported cancer types were limited to breast, cervical, colorectal, and prostate cancer [17]. However, the previously reported practice rate of cancer screening in cancer survivors does not provide insight on whether appropriate cancer screening for cancer survivors is performed. This is because there is no standardized cancer screening guideline for cancer survivors. In the USA, cancer screening guidelines for cancer survivors are based on expert consensus. It is difficult to recommend those guidelines or apply them to the NCSP because they are not evidence-based [18]. Therefore, cancer screening for cancer survivors is dependent on their physician’s opinion. In Korea, there are no cancer screening guidelines for cancer survivors. The NCSP participation rate among cancer survivors is based on the cancer screening guidelines for the general population.

After 2002, the participation rate in the NCSP increased in Korea until 2014; however, Korea’s participation rate in the NCSP was relatively lower than in other countries. This low level of participation in the NCSP suggests a health-system-related characteristic as the cause [19]. Those eligible for the NCSP can visit any of the certified screening units based on the information from their invitation letter. This means that their decision to take part in the NCSP is voluntary and without any advice. Therefore, considering the major role of general practitioners or family doctors in increasing participation rates [20,21,22], improving the perception of cancer screening in the medical staff of a community has been suggested to increase participation.

The participation rate of stomach cancer screening among cancer survivors increased from 6.8% in 2002 to 50.3% in 2014. In addition, the participation rates in colorectal, breast, and cervical cancer screening increased. The participation rate of each cancer screening type, including stomach, breast, and cervical cancer, was more than 45%. However, it was recently 28% in colorectal cancer screening. These participation rates are lower than those in the USA [23,24,25]. The increasing participation rate of stomach cancer screening for the last 13 years may be due to the improvement in accessibility to certified gastric cancer screening units. The National Health Insurance Service (NHIS) designated gastric cancer screening units from 1830, in 2007, to 2418, in 2009 [26]. In Korea, the pattern of the participation rate in the NCSP for both cancer survivors and the general population was similar when compared according to the economic status of the populations. The participation rate for NCSP in the poor economic status population was lower than that in the higher economic status population among cancer survivors. This difference in economic status for cancer screening practice was found in a previous study performed in Japan [27].

Cancer survivors with more time since their diagnosis were more likely to participate in the NCSP in Korea. Generally, the participation rates of the NCSP were high in cancer survivors over five years after diagnosis for all types of cancer screening, except for cervical cancer. The participation rate of cervical cancer screening dropped in 2005 due to the policy changes. Before 2005, people took part in a cervical cancer screening program through the health checkup service of the NHIS. After 2005, the cervical cancer screening program was provided by the NCSP [19]. These results were not consistent with those of a previous study in that the practice rate of cancer screening within five years after diagnosis was low for cancer survivors [28]. The high cancer screening practice rate in cancer survivors within five years after diagnosis is caused by fear of recurrence. Moreover, in Korea, the follow-up period after cancer treatment is generally recommended as five years. Cancer survivors come for more follow-up visits at the hospital during the first three years than in the fourth or fifth years [29]. During the follow-up period, cancer survivors receive a checkup to identify cancer recurrence and care for other symptoms after treatment. This indicates that cancer survivors have an opportunity to receive survivorship care from oncologists or primary care physicians during this period. However, cancer survivors may feel anxious when transitioning from being cancer patients to general individuals due to the patients’ fear of recurrence [30]. Therefore, the higher cancer screening rates in long-term cancer survivors can be explained in terms of psychosocial characteristics and changes in the cancer care environment. In other words, the perception of cancer screening participation as a self-care strategy related to cancer recurrence or a second primary cancer emerged under the previously mentioned context.

The participation rates in the NCSP among cancer survivors in Korea indicate a high demand for cancer screening. Most experts agreed that surveillance for recurrence and primary cancer screening are needed [18]. However, there are certain limitations to this study. First, there are no evidence-based cancer screening guidelines for cancer survivors in Korea. To address this, more information about the benefit and harm of screening cancer survivors is needed. Second, the results only described the participation rate of the NCSP among cancer survivors. Thus, we cannot conclude whether the effects of the NCSP can be applied to cancer survivors. Third, it included only NCSP participants and did not consider the population who underwent cancer screening in the private sector.

## 4. Materials and Methods

The NCSP protocol recommends screening for all Korean men and women (Table 4). A detailed description of the NCSP is given elsewhere [19]. The NHIS covered the whole population of Korea. The NHIS selected eligible men and women for each cancer type based on cancer screening protocols issued by the NCSP in Korea. As part of the NCSP, all eligible men and women received an invitation letter, along with information on screening methods, an available period for screening, and the location of screening units, from the NHIS, beginning in January of each year. Eligible people who received an invitation letter voluntarily visited a certified screening unit after an appointment. According to the socioeconomic status, insurance status was classified into one of three categories: medical aids program (MAP) recipients (extremely poor people who received livelihood assistance and were unable to pay for health care or insurance), NHIS beneficiaries of low-income status (target population for free-of-charge screening), and NHIS beneficiaries of high-income status (target population for screening with a copayment). The target population of the NCSP was categorized into the general population and cancer survivors, exclusively. Cancer survivors were defined as those registered for any cancer in the Korea Central Cancer Registry by December 31 of the year prior to being included in the target population of the NCSP. We ascertained where the study population was screened using the NCSP database since 2002.

In this study, the NCSP for liver cancer was not considered because of the surveillance for high-risk groups. The participation rates for the NCSP were calculated as the percentage of people who participated in each of the four cancer screening programs independently and analyzed on a single-year basis. To estimate changes in participation rates, we assessed the APC by comparing the rates for 2002 and 2014 as relative rates. These rates are reported as the average APC with 95% CIs. All statistical analyses were conducted using SAS software, version 9.4 (SAS Inc., Cary, NC, USA). This study was approved by the Institutional Review Board of the National Cancer Center in Korea (approval number: NCCNCS-08-129).

## 5. Conclusions

Cancer survivors with fear of cancer recurrence take part in cancer screening as a self-care strategy. There are no evidence-based cancer screening guidelines for cancer survivors. In Korea, there is no report on how many cancer survivors participate in the NCSP. This study reported the NCSP participation rates among cancer survivors in Korea. The results show a high demand for cancer screening among cancer survivors. Therefore, we suggest further research to develop appropriate evidence-based cancer screening guidelines for cancer survivors.

## Figures and Tables

**Figure 1 cancers-13-00081-f001:**
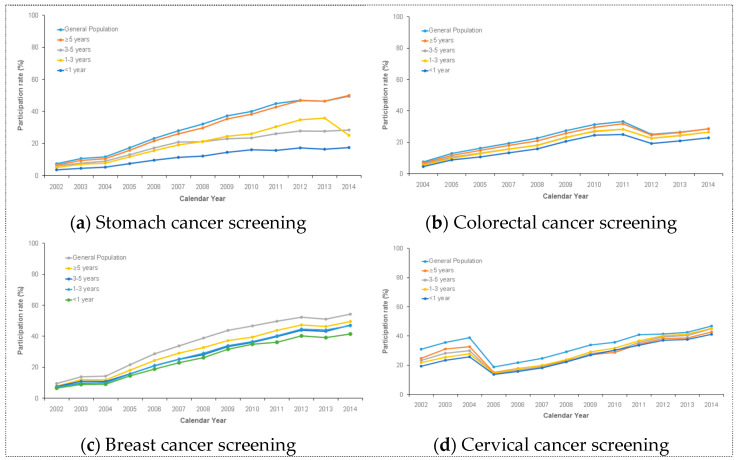
This figure shows the increasing trend in NCSP participation among cancer survivors for over 10 years. Specifically, this shows the trends in (**a**) stomach cancer screening, (**b**) colorectal cancer screening, (**c**) breast cancer screening, and (**d**) cervical cancer screening.

**Table 1 cancers-13-00081-t001:** Target population size of the National Cancer Screening Program (NCSP) by the general population and cancer survivors.

Year	Stomach Cancer Screening		Colorectal Cancer Screening		Breast Cancer Screening		Cervical Cancer Screening	
	General Population	Cancer Survivors	General Population	Cancer Survivors	General Population	Cancer Survivors	General Population	Cancer Survivors
2002	9,485,018	200,207	-	-	4,688,736	111,544	5,012,237	107,875
2003	9,395,771	223,721	-	-	4,846,475	128,171	4,973,196	121,247
2004	10,235,512	257,864	5,650,571	198,688	5,049,584	144,247	5,504,964	141,664
2005	10,347,743	298,721	5,929,189	235,668	5,260,514	169,128	6,085,564	176,436
2006	12,024,044	376,222	6,860,621	295,802	5,939,202	209,778	7,108,745	221,761
2007	11,655,452	404,151	6,806,364	320,942	5,897,940	229,974	6,900,126	241,182
2008	12,978,839	488,396	7,633,719	388,996	6,442,645	277,029	7,723,325	293,726
2009	12,852,587	534,858	7,796,348	433,615	6,520,344	309,398	7,618,099	325,501
2010	12,054,260	553,071	8,339,492	499,093	5,952,877	318,409	7,243,846	342,329
2011	12,283,016	630,525	8,568,200	560,726	6,446,440	372,829	7,887,116	392,538
2012	11,912,694	656,771	14,265,586	1,037,533	6,050,919	381,164	7,979,575	413,460
2013	11,801,373	705,399	14,212,644	1,102,351	6,105,802	418,707	7,873,469	450,630
2014	12,059,692	761,793	14,510,135	1,182,681	6,181,827	448,989	7,851,708	480,384

Target population of the NCSP included the general population and cancer survivors; the general population and cancer survivors are mutually exclusive.

**Table 2 cancers-13-00081-t002:** Trends in participation rates (%) of the NCSP among cancer survivors and the general population.

Cancer Screening Type	Calendar Year													APC (95% CI)	
	2002	2003	2004	2005	2006	2007	2008	2009	2010	2011	2012	2013	2014		
Stomach cancer															
Cancer survivors	6.8	9.8	11.1	16.6	22.6	27.1	30.8	36.3	39.2	43.5	47.5	47.1	50.3	11.5	(8.2, 14.9)
General population	7.5	10.7	11.8	17.5	23.4	28.2	32.5	37.6	40.4	45.3	47.3	46.7	49.9	12.3	(8.6, 16.1)
Colorectal cancers															
Cancer survivor			6.6	11.5	14.6	17.7	20.5	25.5	29.3	31.1	24.2	25.8	28.0	6.1	(0.8, 11.7)
General population			7.5	12.9	16.2	19.4	22.7	27.4	31.3	33.3	25.1	26.5	28.6	6.5	(0.9, 12.5)
Breast cancer															
Cancer survivors	7.9	11.6	11.9	18.1	24.2	28.8	32.5	37.5	40.1	43.9	47.7	46.9	49.7	10.4	(7.2, 13.7)
General population	9.5	13.9	14.3	21.7	28.7	33.8	38.9	43.8	46.7	49.7	52.3	51.1	54.3	10.6	(7.0, 14.4)
Cervical cancer															
Cancer survivors	23.2	28.8	31.0	15.9	19.0	21.3	25.1	29.9	32.0	37.3	40.1	41.4	45.2	7.2	(4.3, 10.1)
General population	31.0	36.6	38.9	18.8	21.8	24.8	29.2	33.9	35.7	40.9	41.4	42.6	46.9	4.0	(0.9, 7.2)

APC, annual percent change; CI, confidence interval.

**Table 3 cancers-13-00081-t003:** Trends in participation rates (%) of the NCSP among cancer survivors and the general population by economic status.

Cancer Screening Type		Calendar Year													APC (95% CI)	
		2002	2003	2004	2005	2006	2007	2008	2009	2010	2011	2012	2013	2014		
Stomach cancer																
NHIS, upper	Cancer survivors	4.2	7.0	8.4	12.5	23.1	29.4	32.6	28.1	40.7	42.6	48.6	46.2	52.3	11.7	(7.2, 16.5)
	General population	5.4	8.4	9.7	12.4	25.3	31.6	34.9	40.2	43.2	45.5	50.2	47.5	53.9	13.2	(8.1, 18.6)
NHIS, lower	Cancer survivors	11.4	15.0	18.2	22.2	24.1	27.7	32.0	36.3	40.9	46.3	47.8	49.2	49.6	9.5	(7.2, 11.9)
	General population	11.4	14.7	16.9	21.2	22.8	26.8	31.8	36.7	39.6	46.1	45.5	46.7	47.3	10.3	(7.8, 12.9)
MAP	Cancer survivors	8.7	9.5	10.2	14.0	13.8	15.9	17.5	22.5	23.6	32.2	35.0	37.2	37.9	14.1	(12.2, 16.1)
	General population	9.9	10.2	11.0	14.7	15.1	17.7	19.4	24.4	25.1	34.3	35.6	37.5	38.2	13.4	(11.6, 15.1)
Colorectal cancer																
NHIS, upper	Cancer survivors			4.7	7.8	14.1	18.6	21.2	26.2	29.0	30.0	22.5	23.6	27.4	5.5	(−1.0, 12.5)
	General population			5.6	9.3	16.4	20.6	23.7	28.7	31.7	32.2	24.0	24.9	28.8	6.5	(−0.6, 14.1)
NHIS, lower	Cancer survivors			11.4	16.0	16.3	18.8	22.1	26.9	31.8	35.0	26.4	28.1	29.0	4.8	(0.0, 9.8)
	General population			12.0	16.7	17.1	19.7	23.5	28.1	32.8	36.3	26.3	28.0	28.7	5.2	(0.3, 10.4)
MAP	Cancer survivors			6.0	12.1	10.4	11.7	12.6	17.5	20.3	21.2	23.2	24.6	25.1	11.4	(8.2, 14.6)
	General population			6.2	12.1	10.7	12.4	13.3	18.1	20.8	21.7	23.1	24.0	25.2	11.3	(8.1, 14.7)
Breast cancer																
NHIS, upper	Cancer survivors	4.7	7.7	9.3	13.9	24.8	31.1	34.3	38.9	40.1	42.3	48.4	45.0	50.9	10.6	(6.1, 15.3)
	General population	6.4	10.3	12.0	18.1	31.5	37.9	42.0	45.8	49.4	48.9	54.6	50.5	57.1	11.1	(5.9, 16.6)
NHIS, lower	Cancer survivors	11.9	18.3	18.3	26.9	25.8	29.7	34.0	39.1	42.7	47.2	48.4	49.5	49.8	8.6	(6.3, 10.9)
	General population	13.4	19.5	19.6	26.1	28.3	32.6	38.7	43.5	47.0	51.8	51.5	52.5	53.2	9.1	(6.5, 11.6)
MAP	Cancer survivors	13.1	12.1	11.4	15.4	15.5	17.3	19.4	24.1	25.6	33.9	36.7	38.3	38.4	12.3	(10.3, 14.3)
	General population	15.0	13.0	12.3	16.4	17.4	20.1	22.4	27.5	28.4	37.2	38.2	39.5	40.0	11.4	(9.4, 13.4)
Cervical cancer																
NHIS, upper	Cancer survivors	23.3	28.8	32.1	15.3	21.4	24.4	28.0	32.2	33.5	36.8	43.0	41.0	47.6	6.4	(3.9, 9.1)
	General population	31.8	36.0	40.1	19.8	26.8	29.9	33.5	37.5	38.8	43.0	44.4	43.5	50.9	3.5	(1.1, 6.0)
NHIS, lower	Cancer survivors	29.1	40.0	41.6	17.8	18.7	21.1	25.2	30.4	33.4	39.2	40.2	42.8	44.1	5.9	(2.0, 9.8)
	General population	35.6	44.8	47.3	19.0	19.7	22.8	27.9	33.0	35.4	40.3	39.7	42.6	44.7	3.3	(−0.9, 7.7)
MAP	Cancer survivors	13.9	10.2	9.4	12.2	11.1	11.7	13.0	17.6	19.1	26.5	29.0	31.3	32.5	12.3	(9.0, 15.7)
	General population	16.1	11.0	10.4	13.0	12.2	13.1	14.6	19.2	20.7	28.2	29.4	31.2	32.1	9.8	(6.4, 13.3)

APC, annual percent change; CI, confidence interval; NHIS, National Health Insurance Service; MAP, Medical Aid Program.

**Table 4 cancers-13-00081-t004:** Cancer screening protocols issued by the NCSP in Korea.

Cancer	Target Population	Interval (yr)	Test
Stomach	Aged ≥ 40 years	2	Upper endoscopy or upper gastrointestinal series
Colorectal	Aged ≥ 50 years	1	Fecal occult blood test (FOBT) ^1^
Breast	Women aged ≥ 40 years	2	Mammography
Cervical	Women aged ≥ 20 years	2	Pap smear

^1^ In the case of an abnormality on FOBT, colonoscopy or a double-contrast enema is recommended.

## Data Availability

Restriction apple to the availability of these data. Data was obtained from NHIS, Korea.

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
