# Peer review of "Trends in Participation Rates of the National Cancer Screening Program among Cancer Survivors in Korea"

_cancers, 2020, doi:10.3390/cancers13010081_

Round 1
Reviewer 1 Report
Interesting paper.
I have some comments/suggestions:
a) Table 2 shows the remarkable increase of both general population and cancer survivors compliance over time to the NCSP, unfortunately remaining too low according to the international standards for cost/benefit of screening for any cancer. Thus Authors should in discussion stress it and suggest more and more improvement of screening adhesion
b) I suppose stomach cancer screening started because of a very high incidence in Korea.Actually it seems to me it is ongoing in Korea, Japan, and New Zealand (on Maori high risk population only). Authors should mention in Introduction incidence of stomach cancer in the Country
c) One other important observation is the reported role of economic status in screening participation, also improved over time and showing a high compliance in low economic status patients. It is important for reader to know if “NHIS” and “MAP” coverage regards any Korean citizen or if there is some “excluded”. Furthermore, the world data bank (https://data.worldbank.org) reports Out-of-Pockety money payed by Korean citizens was 206 USD in 2002 and 769 USD in 2017. Authors should mention and comment these data
d) Usually in a paper there is Introduction, Materials and Methods, Results, Discussion, Conclusions, References.
In the submitted paper the order is:Introduction (very short), Results, Discussion, Materials and Methods, Conclusions.
Authors should reorder the chapter as usual
e) Tables, wich are included in the text, are not “Supplementary material”
Author Response
Response to Reviewer 1 Comments
Point 1. Table 2 shows the remarkable increase of both general population and cancer survivors compliance over time to the NCSP, unfortunately remaining too low according to the international standards for cost/benefit of screening for any cancer. Thus Authors should in discussion stress it and suggest more and more improvement of screening adhesion.
Response 1. As your comments, sentence was added in LINES 135-141 as followed: “However, the participation rate of NCSP was relative lower than other countries. The reason about low level of participation rate of NCSP suggest a health system related characteristics [13]. Eligilble people for NCSP can visit any of certified screening units based on the information from invitation letter. This means that the decision to take part in NCSP is voluntarily without any advice. Therefore, considering the major role of general practitioners or family doctors in increasing participation rates [14-15], it suggests the improving perception of cancer screening in medical staff in community.
Point 2. I suppose stomach cancer screening started because of a very high incidence in Korea. Actually it seems to me it is ongoing in Korea, Japan, and New Zealand (on Maori high risk population only). Authors should mention in introduction incidence of stomach cancer in the Country.
Response 2. As your comments, incidence of stomach cancer was added in introduction as followed in LINES 51-53: “Especially, the difference between the cancer screening guideline in Korea and other countries is that stomach cancer screening is recommended because high incidence of stomach cancer in Korea. The stomach cancer ranked the first in 2017, except thyroid cancer, in Korea [6].”
Point 3. One other important observation is the reported role of economic status in screening participation, also improved over time and showing a high compliance in low economic status patients. It is important for reader to know if “NHIS” and “MAP” coverage regards any Korean citizen or if there is some “excluded”. Furthermore, the world data bank (https://data.worldbank.org) reports Out-of-Pockety money payed by Korean citizens was 206 UDS in 2002 and 769 USD in 2017. Authors should mention and comment these data.
Response 3. As your comments, the definition of “NHIS” and “MAP” was added in material and method section as followed: “In the NCSP, all eligible man and women received an incitation letter, along with information on screening methods and the location of screening units, from the National Health Insurance Service (NHIS), beginning in January of each year. According to the socioeconomic status, insurance status was classified into one of three categories: medical aids program (MAP) recipients (extremely poor people who received livelihood assistance and were unable to pay for health care or insurance), NHIS beneficiaries of low-income status (target population for free-of-charge screening), and NHIS beneficiaries of high-income status (target population for screening with a copayment).”
Point 4. Usually in a paper there is Introduction, Materials and Methods, Results, Results, Discussion, Conclusions, References.
In the submitted paper the order is: Introduction (very short), Results, Discussion, Materials and Methods, Conclusions.
Authors should reorder the chapter as usual.
Response 4. According to the instruction for Authors in cancers, the order of research manuscript section is introduction, results, discussion, materials and methods, and conclusion. (Cancers | Instructions for Authors (mdpi.com))
Point 5. Tables, which are included in the test, are not “Supplementary material”
Response 5. As your comments, “Supplementary material” was corrected.
Reviewer 2 Report
The focus of this paper is interesting – cancer survivors being generally excluded from screening programmes under the assumption that they adequately followed up according to (local, national, international) guidelines. However, such assumption is not necessarily true. However, in the current form the results of the paper are hardly generalizable and comparable with the figures from other Countries. Fundamental context information is missing: what do the Authors precisely mean with “Screening Programmes”? Are there in Korea population based programmes, actively inviting the target population? Are they free of charge? What about accessibility (overall, and along the study period)? Why do the Authors refer to participation of cancer survivors as “self-care strategy” – how can this be compared with participation to a “screening programme”?
Comments
- The main message that is conveyed in different parts of the paper is that colorectal cancer survivors participate less than survivors from the other cancers. In fact, the participation of colorectal cancer survivors is in line with the participation to colorectal screening of the general population, and the same applies to the other cancers. The text should be modified, in order to convey the main message that the general population shows different participation to the different screenings, and that cancer survivors behave in the same way than the general population. Therefore, please report the participation of the general population in the abstract before that of cancer survivors – LINES 27 and 28. Similarly, the first figures to report in the Results Section (LINE 74, LINES 91 and 92) are the participation rates of the general population, followed by those of cancer survivors.
- I would find extremely informative if participation were stratified according to the number of years from cancer diagnosis. This would answer the question: what is the participation of cancer patients whose cancer was diagnosed recently (e.g., in the last 2-5 years) as compared with those who developed a cancer before?
- I simply do not understand the meaning of the sentence starting with “however, …” in LINE 84.
- In different parts of the paper the Authors state that Korea has not produced any cancer screening guidelines for cancer survivors. However in LINES 50-51 they state the opposite. Please clarify.
- Table 1. For each cancer, I suggest to report the general population in the first column, the number of cancer survivors in the second column, and to add near cancer survivors the corresponding percentage of the general population (within round parenthesis).
- Figure 1, d). What determined the drop in participation in 2005? It seems unlikely that such a drop is related to a real reduction in participation.
- A sentence is repeated in LINES 117 and 118
- It is necessary that the Authors describe in detail the characteristics of the National Cancer Screening Programme(s). In many Countries, screening programmes actively invite (e.g., by mail) the target population to uptake the screening test. Is this the case in Korea? Since the Authors talk about “self-care strategy” (LINE 187), it does not seem so. Moreover, can the Authors describe whether the accessibility to screening was stable along the study period, or if it changed during the study? Are the screening tests free of charge?
- LINE 122: the Authors state that the study was based on insurance data. This sentence should be moved in the Methods Section and more information should be added in order to help the readers to understand what specific data were used. Were such data homogeneously available during the whole study period?
- In general, the Methods section should be improved. Please describe how the exams performed as follow up from were disentangled from those performed for “screening”. Also clarify how the economic categories were defined.
- LINE 136: What determined the increase in participation to stomach screening from 2002 to 2014?
- In LINE 140 the Authors compare their results for colorectal cancer screening with those in the US. Given that the FOBT is used in Korea, while colonoscopy is the screening test in the US, the two programs seem not directly comparable. The Authors could consider comparing their figures with those of Countries where the FOBT is used instead.
- LINES 152 and 153: please describe what exams are performed for follow up by cancer survivors.
- LINE 193. The titles of tables and Figures are not Supplementary Materials!
Author Response
Response to Reviewer 2 Comments
Point 1. The main message that is conveyed in different parts of the paper is that colorectal cancer survivors participated less than survivors from the other cancers. In fact, the participation of colorectal cancer survivors is in line with the participation to colorectal screening of the general population, and the same applies to the other cancers. The text should be modified, in order to convey the main message that the general population shows different participation to the different screenings, and that cancer survivors behave in the same way than the general population. Therefore, please report the participation of the general population in the abstract before that of cancer survivors – LINES 27 and 28. Similarly, the first figures to report in the Results Section (LINE 74, LINES 91 and 92) are the participation rates of the general population, followed by those of cancer survivors.
Response 1. “The participation rates of patients with stomach, breast, and cervical cancer were higher than 40%” was mispresented. Cancer survivors were not categorized by cancer diagnosis. We just identified how many cancer survivors take part in the national cancer screening program. Therefore, sentence what you are pointed out rewrite as followed: “The participation rated of general population and cancer survivors were higher than 40% in stomach, breast, and cervical cancer screening.” (LINES 27 and 28)
The results section (LINE 74, LINES 91 and 92) also rewrites as your comments as followed.
- Before revision (LINE 74) “Among cancer survivors, the participation rate of patients in the NCSP in 2002-2014 increased for all cancer types. However, the magnitudes of the increase varied between cancer types. In 2014, the participation rates of patients with stomach, breast, and cervical cancer were higher than 40%, and they were reportedly higher than that of patients with colorectal cancer by 28%.”
- After revision (LINE 76) “Among general population and cancer survivors, the participation rate in the NCSP IN 2002-2014 increased for all cancer screening types. However, the magnitudes of the increase varied between cancer screening types. In 2014, the participation rated of general population and cancer survivors were higher than 40%, and these rates were reportedly higher than that of colorectal cancer screening.”
- Before revision (LIENS 91 and 92) “However, it has increased in the last 13 years in both cancer survivors and the general population (Table 3). In 2014, the lowest NCSP participation rate was observed among all cancer survivors in the lower economic status group. This observation was consistent in the general population as well.”
- After revision (LINES 92-95) “However, it has increase in the last 13 years in both the general population and cancer survivors (Table 3). In 2014, the lowest NCSP participation rate was observed among general population in the lower economic status group. This observation was consistent in cancer survivors as well.”
Point 2. I would fine extremely informative if participation were stratified according to the number of years from cancer diagnosis. This would answer the question: what is the participation of cancer patients whose cancer was diagnosed recently (e.g., in the last 2-5 years) as compared with those who developed a cancer before?
Response 2. The last part in results section showed the trends in the participation rates of NCSP by group based on time since diagnosis among cancer survivors. In this part, the participation rates of cancer survivors who are categorized by at less than one year after diagnosis, 1-3 years after diagnosis, 3-5 years after diagnosis, and more than 5 years diagnosis was reported.
Point 3. I simply to not understand the meaning of the sentence starting with “however, …” in LINE 84. In different parts of the paper the Authors state that Korea has not produced any cancer screening guidelines for cancer survivors. However in LINES 50-51 they state the opposite. Please clarify.
Response 3. There was mispresented in LINES 50-51 and LINE 84. As your comments, mispresented sentence was corrected.
- Before revision (LINES 50-51) “In many countries, cancer screening guidelines for the general population exist [7]; however, only few countries, including Korea, have cancer screening guidelines for cancer survivors.”
- After revision (LINES 50-51) “In many countries, cancer screening guidelines for the general population exist [7]; however, many countries, including Korea, have no cancer screening guidelines for cancer survivors”
- Before revision (LINE 84) “However, the magnitude of the increasing trend in participation rates among cancer survivors was not observed among patients with cervical cancer.”
- After revision (LINE 85-87) “However, the magnitude of the increasing trend in participation rates among cancer survivors was higher than the among general population in cervical cancer screening.”
Point 4. Table 1. For each cancer, I suggest to report the general population in the first column, the number of cancer survivors in the second column, and to add near cancer survivors the corresponding percentage of the general population (within round parenthesis).
Response 4. Table 1. was corrected as your comments.
Point 5. Figure 1, d). what determined the drop in participation in 2005? It seems unlikely that such a drop is related to a real reduction in participation.
Response 5. According to the study of trends in participation rates for the NCSP in Korea, the reason about a significant drop in participation rate for cervical cancer screening in 2005 is the policy changes in the cervical screening program. From 1988 to 2004, cervical cancer screening was provided through a NHIS health checkup service. In 2005, the cervical cancer screening program was separated from the checkup service and included in the NCSP. This change likely generated confusion among women who had previously underwent cervical cancer screening through the NHIS health checkup service. Furthermore, in the NCSP, invitees must voluntarily decide whether to make a screening appointment or not, while the NHIS health checkup service strongly promoted and encouraged participation. [reference. Suh, M/; Song, S.; Park, B.; Jun J.K.; Choi, E.; Kim, Y.; Choi, K.S. Trends in participation rates for the national cancer screening program in Kroea, 2002-2012. Cancer Res Treat 2017, 49(3), 798-806. DOI:10.4143/crt.2016.186]
Point 6. A sentence is repeated in LINES 117 and 118.
Response 6. Repeated sentences in LINES 117 and 118 was corrected as your comments.
Point 7. It is necessary that the Authors describe in detail the characteristics of the National Cancer Screening Programme(s). In many Countries, screening programmes actively invite (e.g., by mail) the target population to uptake the screening test. Is this the case in Korea? Since the Authors talk about “self-care strategy” (LINE 187), it does not seem so. Moreover, can the Authors describe whether the accessibility to screening was stable along the study period, or is it changed during the study? Are the screening tests free of charge?
Response 7. As your comments, the characteristics of the National Cancer Screening Programme(s) was added in detail in the material and methods section. In Korea, NCSP sent a invitation letter to the target population for providing the information about NCSP and encouraging to take part in the NCSP.
According to the socioeconomic status, insurance status was classified into one of three categories: medical aids program (MAP) recipients (extremely poor people who received livelihood assistance and were unable to pay for health care or insurance), NHIS beneficiaries of low-income status (target population for free-of-charge screening), and NHIS beneficiaries of high-income status (target population for screening with a copayment). This information also added in the material and method section.
Point 8. LINE 122: the Authors state that the study was based on insurance data. This sentence should be moved in the Methods Section and more information should be added in order to help the readers to understand what specific data were used. Were such data homogeneously available during the whole study period?
Response 8. The more information about insurance data was added in material and method section as your comments.
- Added sentences. “In the NCSP, all eligible man and women received an incitation letter, along with information on screening methods and the location of screening units, from the National Health Insurance Service (NHIS), beginning in January of each year. According to the socioeconomic status, insurance status was classified into one of three categories: medical aids program (MAP) recipients (extremely poor people who received livelihood assistance and were unable to pay for health care or insurance), NHIS beneficiaries of low-income status (target population for free-of-charge screening), and NHIS beneficiaries of high-income status (target population for screening with a copayment).”
Point 9. In general, the Methods section should be improved. Please describe how the exams performed as follow up from were disentagled from those performed for “screening”. Also clarify how the economic categories were defined.
Response 9. As your comments, the information about NCSP was added in material and methods section.
Point 10. LINE 136: What determined the increase in participation to stomach screening from 2002 to 2014?
Response 10. The possible explanation for the increase in participation to stomach cancer screening from 2002 to 2014 may be the improvement in accessibility to endoscopy testing. After introduction of the nationwide gastric cancer screening program, a larger number of physicians began to perform endoscopy test to meet the needs of endoscopic screening services. In 2008, there were >4,000 board-certified endoscopic specialists in Korea. Furthermore, to improve accessibility to gastric cancer screening, the NHIS only required clinics and hospitals to possess endoscopic equipment for designation as a gastric cancer screening unit. Whit this change, the number of gastric cancer screening units designated by the NHIS increased from 1,830 in 2007 to 2,418 to 2009. (Reference. Lee, S; Jun, J.K.; Suh, M.; Park, B.; Noh, D.K.; Jung, K.W.; Choi, K.S. Gastric cancer screening uptake trends in Korea: Results for the National Cancer Screening Program from 2002 to 2011. Medicine 2015, 94(8), e533. DOI:10.1097/MD.0000000000000533)
Point 11. In LINE 140 the Authors compare their results for colorectal cancer screening with those in the US. Given that the FOBT is used in Korea, while colonoscopy is the screening test in the US, the two programs seem not directly comparable. The Authors could consider comparing their figure with those of Countries where the FOBT is used instead.
Response 11. As your comments, recommendation for colorectal screening method between Kore and US is different. However, Mayer et al. (2007) reported that 84.6% of cancer survivors reported ever having a colonoscopy or sigmoidoscopy or FOBT, and Clarke et al. (2012) also reported that adherence rate screening among US cancer survivors was more than 40%. Basic methods for colorectal cancer screening is FOBT in US and Korea. In this context, we consider that 28% in colorectal cancer screening was lower than that of US.
Point 12. LINES 152 and 153: please describe what exams are performed for follow up by cancer survivors.
Response 12. As your comments, sentence was added as followed: “During follow-up period, cancer survivors receive a checkup for identifying cancer recurrence and other symptom care after treatment.”
Point 13. LINE 193: The titles of tables and figures are not Supplementary Materials!
Response 13. As your comments, “Supplementary materials” was corrected.
Round 2
Reviewer 1 Report
Interesting and usefull paper
Author Response
Point 1. Comments and Suggestions for Authors. Interesting and useful paper
Response 1. Thank you for your comments.

Reviewer 2 Report
The Authors adequately answered to all my comments. I have just few further requests.
Table 1. Thank you for modifying the Table, please further add near cancer survivors the corresponding percentage of the general population (maybe within round parenthesis)
[e.g. 9,485,018 200,207 (2.11%)]
Figure 1, d). Thank you for explaining what determined the drop in participation in 2005. However, this should be (briefly) reported in the discussion.
Methods: some additional information about the NCSP should be included in the Methods. In detail, please specify the contents of the “invitation letter”: does it include a pre-fixed appointment? Or does it generically invite to contact some facilities of the National Health System in order to fix an appointment?
Also, please clearly state that cancer survivors are not excluded from receiving the invitation letter.
Please state whether insurance data were homogeneously available during the whole study period
Thank you for explaining what determined the increase in participation to stomach screening from 2002 to 2014. However, this should be (briefly) reported in the discussion. Similarly, other variables that could explain the observed trends during the study period should be reported in the discussion and commented.
Author Response
Response to Reviewer 2 Comments (Round 2)
Point 1. Table 1. Thank you for modifying the Table, please further add near cancer survivors the corresponding percentage of the general population (maybe within round parenthesis)
Response 1. The National Health Insurance Service (NHIS) covered whole population in Korea. The NHIS selects eligible men and women for each cancer type based on cancer screening protocols issued by the NCSP in Korea. The target population of the NCSP included the general population and cancer survivors and was categorized into the general population and cancer survivors, exclusively. Cancer survivors were defined as those registered for any cancer in the Korean Central Cancer Registry by December 31 of the year prior to being included in the target population of the NCSP. As your comments, the explanation about the target population of the NSCP was added in material and methods section and Table 1.
- LINES 188-190. “The NHIS covered whole population in Korea. The NHIS selects eligible men and women for each cancer type based on cancer screening protocols issued by the NCSP in Korea.”
- LINES 198-199. “The target population of the NCSP was categorized into the general population and cancer survivors, exclusively.”
Point 2. Figure 1, d). Thank you for explaining what determined the drop in participation in 2005. However, this should be (briefly) reported in the discussion.
Response 2. As your comments, the explanation what determined the drop in participation in 2005 was added in the discussion as followed.
- LINES 159-162. “The participation rate of cervical cancer screening dropped in 2005 due to the policy changes. Before 2005, people took part in a cervical cancer screening program through the health checkup service of a NHIS. After 2005, the cervical cancer screening program was provided by the NCSP [19].”
Point 3. Methods: some additional information about the NCSP should be included in the Methods. In detail, please specify the contents of the “invitation letter”: does it include a pre-fixed appointment? Or does it generically invite to contact some facilities of the National Health System in order to fix an appointment?
Response 3. As your comments, the contents of the “invitation letter” was added as followed.
- LINES 190-193. “In the NCSP, all eligible men and women receive an invitation letter, along with information on screening methods, an available period for screening, and the location units, from NHIS, beginning in January of each year. Eligible people who got an invitation letter visit a certified screening unit after an appointment, voluntarily.
Point 4. Also, please clearly state that cancer survivors are not excluded from receiving the invitation letter.
Response 4. The target population of the NCSP was categorized into the general population and cancer survivors, exclusively in this study. It means that cancer survivors who meet an eligible criteria for cancer screening could got an invitation letter. As your comments, the target population of the NCSP including defined in detail in material and methods section.
Point 5. Please state whether insurance data were homogeneously available during the whole study period.
Response 5. The answer about your comment is “Yes”. Insurance data were homogeneously available during the whole study period.
Point 6. Thank you for explaining what determined the increase in participation to stomach screening from 2002 to 2014. However, this should be (briefly) reported in the discussion. Similarly, other variables that could explain the observed trends during the study period should be reported in the discussion and commented.
Response 6. As your comments, the explanation what determine the increase in participation to stomach cancer screening from 2002 to 2014 was added in the discussion as followed.
- LINES 148-151. “The increasing participation rate of stomach cancer screening for last 13 years may be the improvement in accessibility to certified gastric cancer screening units. The NHIS designated gastric cancer screening units from 1,830 in 2007 to 2,418 in 2009 [26].”